

# Vegetation restoration improved aggregation stability and aggregated-associated carbon preservation in the karst areas of Guizhou Province, southwest China

Hui Yang, Hui Long, Xuemei Li, Xiulong Luo, Yuanhang Liao, Changmin Wang, Hua Cai and Yingge Shu

Guizhou University, College of Agronomy, Guiyang, Guizhou, China

Corresponding author
Yingge Shu, maogen958@163.com

## ABSTRACT

**Background**. The change in the soil carbon bank is closely related to the carbon dioxide in the atmosphere, and the vegetation litter input can change the soil organic carbon content. However, due to various factors, such as soil type, climate, and plant species, the effects of vegetation restoration on the soil vary. Currently, research on aggregate-associated carbon has focused on single vegetation and soil surface layers, and the changes in soil aggregate stability and carbon sequestration under different vegetation restoration modes and in deeper soil layers remain unclear. Therefore, this study aimed to explore the differences and relationships between stability and the carbon preservation capacity (CPC) under different vegetation restoration modes and to clarify the main influencing factors of aggregate carbon preservation.

**Methods**. Grassland (GL), shrubland (SL), woodland (WL), and garden plots (GP) were sampled, and they were compared with farmland (FL) as the control. Soil samples of 0–40 cm were collected. The soil aggregate distribution, aggregate-associated organic carbon concentration, CPC, and stability indicators, including the mean weight diameter (MWD), fractal dimension (D), soil erodibility (K), and geometric mean diameter (GMD), were measured.

**Results**. The results showed that at 0–40 cm, vegetation restoration significantly increased the >2 mm aggregate proportions, aggregate stability, soil organic carbon (SOC) content, CPC, and soil erosion resistance. The >2 mm fractions of the GL and SL were at a significantly greater proportion at 0–40 cm than that of the other vegetation types but the CPC was only significantly different between 0 and 10 cm when compared with the other vegetation types ($P < 0.05$). The >2 mm aggregates showed a significant positive correlation with the CPC, MWD, and GMD ($P < 0.01$), and there was a significant negative correlation with the D and K ($P < 0.05$). The SOC and CPC of all the vegetation types were mainly distributed in the 0.25–2 mm and <0.25 mm aggregate fractions. The MWD, GMD, SOC, and CPC all gradually decreased with increasing soil depth. Overall, the effects of vegetation recovery on soil carbon sequestration and soil stability were related to vegetation type, aggregate particle size, and soil depth, and the GL and SL restoration patterns may be more suitable in this study area. Therefore, to improve the soil quality and the sequestration of organic

carbon and reduce soil erosion, the protection of vegetation should be strengthened and the policy of returning farmland to forest should be prioritized.

## INTRODUCTION

Karst areas are primarily distributed in southwestern China, and due to deforestation and soil erosion, rocky desertification is becoming increasingly serious, soil productivity is declining, and soil degradation has become the greatest threat to the ecological stability, residents, and local economic development of the karst areas (*Zhou et al., 2008*; *Lan, 2021*). In 1999, China implemented the ''Returning Farmland to Forest'' project since vegetation restoration can increase soil nutrients through apoptosis and root exudates and effectively improve soil quality, organic carbon content, and soil erosion through the formation of a stable soil structure (*Liu, Han & Zhang, 2019*; *Dong et al., 2022*). However, there is no agreement on the type of vegetation restoration that is most appropriate for the karst area. For example, *Xiao et al. (2021)* indicated that forage grass is the best vegetation restoration type in karst areas. Whereas *Yu et al. (2022)* suggested that orange orchards had the best recovery pattern. Therefore, there is a need to explore further the applicability of different vegetation types in karst areas.

Soil is the biggest carbon pool in terrestrial ecosystems, so small fluctuations in the soil carbon can affect the atmospheric $CO_2$ concentration, which leads to global climate change (*Lehmann & Kleber, 2015*; *Liu et al., 2020*). Soil organic carbon (SOC) is also one of the main soil properties that affect soil fertility and plant growth, and its loss may lead to a decrease in soil quality (*Briedis et al., 2021*). The SOC reservoir can be indirectly increased by conservation tillage and conservation of forests (*Lal & Bruce, 1999*; *González-Rosado et al., 2023*). Therefore, since the implementation of the ''Returning Farmland to Forest'' project, many studies have been conducted on the effect of vegetation restoration on the soil aggregate stability and carbon reservoir. For instance, *Yu et al. (2022)* showed that afforestation significantly improved the organic carbon content in the soil and aggregates. Moreover, *Zhao et al. (2017)* indicated that paddy fields that were converted to woodland (WL) and dry land resulted in a decrease in the aggregate-associated organic carbon content in southwest China. These contrasting results may be due to various factors, including the soil, recovery time, climate, and vegetation (*Hu & Lan, 2020*; *Lu et al., 2022*).

Other studies found that the soil aggregates, organic carbon content, and aggregate stability increased in the 0–20 cm soil layer after fallow but the soil texture and soil erodibility (K) did not significantly change under different land uses (*Liu, Han & Zhang, 2019*). Furthermore, (*Lu et al., 2022*) found that natural grassland (GL) improved the soil aggregate stability and aggregate-associated SOC concentration when compared with *Caragana* plantations. However, these studies were mainly focused on the soil surface layer (0–20 cm) and a single vegetation type. Thus, there is a need for research that focuses on

different vegetation types and explores the stability and mechanism of change of the soil aggregates and organic carbon in the soil layer.

Soil aggregates are the basic unit of the soil structure, and they affect the nutrients, water, heat, and air in the soil, maintain and stabilize the different soil layers, and influence the dynamics of the soil quality and health (*Six & Paustian, 2014*). Generally, soil aggregates are divided into macro-aggregates (>2 mm), meso-aggregates (0.25–2 mm), and micro-aggregates (<0.25 mm; *Lu et al., 2022*). Organic carbon is an important material for aggregate formation, and high organic carbon content can improve soil aggregate formation (*Yang, Lv & Li, 2015*). Soil aggregates can prevent organic carbon from being mineralized due to physical protection, which is conducive to carbon fixation (*Six et al., 2004*). The growth of plants can introduce a large amount of organic matter into the soil system, which improves soil quality and structure (*Dou et al., 2020*). Therefore, the restoration of vegetation can accelerate the cementation of soil aggregates and redistribute aggregates of different sizes, thus determining the size and direction of soil carbon accumulation (*Qiu et al., 2015*).

Different aggregate sizes can significantly affect the organic carbon content (*Dong et al., 2017*). The aggregate-associated SOC is stored in macro aggregates (>0.25 mm) due to their organic carbon content being stable and in macro micro-aggregates (<0.25 mm) due to their strong physical protection and low carbon turnover (*Gelaw, Singh & Lal, 2015*; *Okolo et al., 2020*). Many studies have found >2 mm aggregates to be the main location of carbon accumulation (*Du et al., 2023*; *Jozedaemi & Golchin, 2024*); (*Jia et al., 2012*). Additionally, some studies indicate that both macroaggregates and microaggregates are the main components of organic carbon sequestration (*Yu et al., 2022*). Since there have been contrasting findings, it is necessary to further clarify the driving role of different particle-size aggregates in organic carbon changes.

Good soil structure depends on the soil aggregate distribution and aggregate stability (*Egan, Crawley & Fornara, 2018*). The number of >0.25 mm aggregate fractions is a reliable indicator of soil quality, and the soil has an excellent soil structure when the proportion of water-stable aggregates that are >0.25 mm is more than 70% (*Lin et al., 2020*). The mean weight diameter (MWD), fractal dimension (D), and geometric mean diameter (GMD) can used to represent the stability of the aggregates. Higher MWD and GMD values indicate good soil structure and high stability (*Nimmo & Perkins, 2002*). However, the K is a key parameter for evaluating the soil erosion sensitivity, which can also reflect the soil quality to some extent (*Wang et al., 2013*). Nevertheless, many studies on aggregate stability and carbon pools have focused on pepper forests with different tillage practices (*Sekaran, Sagar & Kumar, 2021*), different fertilizer application methods (*Du et al., 2017*), and different restoration time sequences (*Lv et al., 2022*). Few studies have investigated the relationships between soil aggregate stability and soil carbon preservation capacity (CPC) while considering multiple vegetation restoration modes. As previous studies have not focused on this relationship, we attempted to explore this aspect.

Consequently, this study was the first to explore the differences between soil aggregate stability and CPC under different vegetation restoration types to clarify the suitable vegetation restoration types in karst areas. Clarifying the effects of four vegetation

**Table 1** **The basic information for the study area.** Farmland (FL), grassland (GL), shrubland (SL), woodland (WL), and garden plots (GP).

| Vegetation types | Longitude | Latitude | Altitude (m) | Main vegetation |
|---|---|---|---|---|
| FL | 106.4800 | 26.3163 | 1,243.06 | maize (*Zea mays*) |
| GL | 106.4862 | 26.3244 | 1,236.36 | *Imperata cylindrica* (L.) P. Beauv |
| GP | 106.5393 | 26.3485 | 1,225.38 | peach trees (*Prunus persica* (L.) Batsch) |
| SL | 106.4783 | 26.3209 | 1,243.06 | *Pyracantha fortuneana* (Maxim.) Li, *Artemisia princeps* Pamp. var. orientalis (Pamp.) Hara, *Rubus idaeus* L. |
| WL | 106.4851 | 26.3221 | 1,223.18 | *Catalpa bungei* C. A. Mey, *Celtis sinensis* Pers |

restoration modes, namely GL, shrubland (SL), WL, and garden plots (GP), on the soil aggregate stability and carbon pools in karst areas can provide a deeper understanding of soil carbon sequestration, which is of great significance for ecological restoration and sustainable soil development. Based on previous studies, we hypothesized that (1) vegetation restoration promotes the formation of macro aggregates (>2 mm), which significantly improves aggregate stability and organic CPC, and that (2) the aggregate-associated CPC is significantly and positively correlated with aggregate stability. We aimed to (1) investigate the distribution characteristics and cluster stability of the soil aggregates under different vegetation restoration modes; (2) analyze the distribution of organic carbon in the aggregates of different grain sizes under different vegetation restoration modes; and (3) clarify the relationship between the cluster stability index of the aggregates of different grain sizes and the organic carbon pool of the aggregates. The findings of this study can provide insights into the carbon conservation capacity of agglomerates, and provide a scientific basis for the ecological management of the southwest karst region.

## MATERIALS & METHODS

### Study area

The study area was in Machang Town, Pingba District, Anshun City, Guizhou Province (26.15′18″–26.37′45″N, 105. 59′24″–106.34′06″E), a mountainous plateau in southwest China, which belongs to the Yunnan-Guizhou Plateau. The climate in this area is humid subtropical with monsoon-like conditions and an average yearly rainfall of 1,165 mm. The temperature remains at around 13.3 °C throughout the year. The elevation is 1,018–1,647 m, and the terrain conditions are typical karst landforms. The rocks are mainly limestone and marl, and the soil type is predominantly limestone soil. In this area, maize (*Zea mays* L.) is the main crop in the cultivated land, and the dominant species in the GL is *Imperata cylindrica* (L.) P. Beauv. Peach trees (*Prunus persica* (L.) Batsch) are the main planting crops in the GP. *Pyracantha fortuneana* (Maxim.) Li, *Artemisia princeps* var. orientalis (Pamp.) Hara, and *Rubus idaeus* L. are the dominant species in the SL. *Catalpa bungei* C. A. Mey and *Celtis sinensis* Pers are the main species in the WL. The basic information for the study area is shown in Table 1 and the locations of the sampling points are shown in Fig. 1.

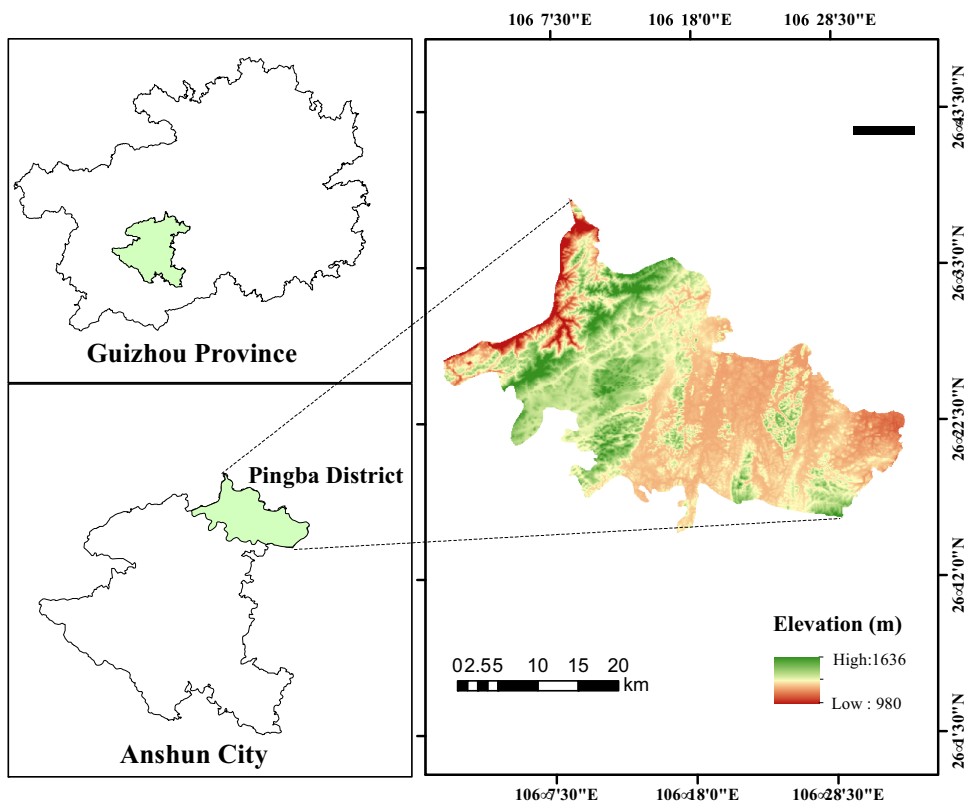

**Figure 1** Location of the study area in the Pingba District, Guizhou, China.

**Table 2** Basic soil characteristics of soil from 0–40 cm for different vegetation restoration types (average ± standard error). Farmland (FL), grassland (GL), shrubland (SL), woodland (WL), and garden plots (GP).

| Vegetation | BD (g/cm³) | pH | AN (mg/kg) | TP (g/kg) | AP (mg/kg) | TK (g/kg) | AK (mg/kg) | SOC (g/kg) |
|---|---|---|---|---|---|---|---|---|
| FL | 1.26 ± 0.02 | 5.48 ± 0.11 | 111.7 ± 15.8 | 0.12 ± 0.01 | 2.5 ± 0.49 | 11.28 ± 0.37 | 57.77 ± 2.62 | 17.16 ± 1.06 |
| GL | 1.08 ± 0.01 | 6.69 ± 0.2 | 202.55 ± 17.84 | 0.16 ± 0.01 | 1.74 ± 1.09 | 13.22 ± 0.64 | 82.25 ± 11.41 | 33.48 ± 1.97 |
| GP | 1.13 ± 0.02 | 6.62 ± 0.11 | 179.56 ± 8.95 | 0.16 ± 0.01 | 2.32 ± 0.63 | 12.57 ± 0.1 | 100.1 ± 6.44 | 28.48 ± 0.84 |
| SL | 1.12 ± 0.1 | 7.07 ± 0.18 | 199.58 ± 27.84 | 0.12 ± 0.02 | 1.01 ± 0.23 | 11.42 ± 1.24 | 70.11 ± 11.72 | 31.94 ± 5.23 |
| WL | 1.2 ± 0.06 | 6.76 ± 0.34 | 217.58 ± 25.1 | 0.15 ± 0.02 | 2.25 ± 0.63 | 16.78 ± 2.3 | 128.31 ± 22.78 | 26.43 ± 4.05 |

## Experimental design

The experimental site was chosen through field investigation and visiting local villagers, and BIGEMAP and Google Earth were used to compare the historical satellite images in a concentrated area with the same slope direction, slope degree, and elevation. In July 2022, different vegetation modes, namely GL, SL, WL, and GP, were selected as sampling sites. The FL was treated as the control in this study. The basic soil conditions are shown in Table 2.

## Soil sample collection

Soil sample collection was conducted in the study area in July 2022. First, the litter and humus layers of the soil surface were removed before collection. Second, soil samples of five soil layers (0–5, 5–10, 10–20, 20–30, and 30–40 cm) in three soil profiles were collected using the "S" type sampling method. The soil sample was put into a box to determine the soil aggregate composition and aggregate-associated organic carbon. The samples were sealed and placed flat to maintain the soil structure during transport. Finally, 75 soil samples were gathered in total and their physicochemical properties were analyzed. In addition, in each soil section, a small ring knife (diameter 5 cm, height 5 cm) was used to cut undisturbed natural soil samples to determine the soil bulk density.

## Determination of the soil physical and chemical indexes
### Separation of the soil aggregates

The dry soil samples were classified into different aggregate size classes by a wet sieving method (*Elliott, 1986*). Initially, a 50 g soil sample was weighed, an XY-100 instrument (Beijing Xiangyuweiye Ltd., China) was used for the wet sieve analysis, and a 2 mm and 0.25 mm sieve separated the aggregation fractions of >2 mm, 0.25–2 mm, and <0.25 mm. The frequency of the instrument was kept between 30–40 r/min, and the oscillation time was 30 min. When the oscillation was completed, the samples were washed on each sieve with deionized water in a large aluminum box. Finally, the aggregates were dried at 55 °C for 24 h and the weight of each component was recorded on weighing paper (accurate to 0.01 g). The soil bulk density was measured using the 100 cm$^3$ ring knife method (*Wang et al., 2022*).

### Determination of the soil chemical indicators

The SOC content was determined by the external heating and potassium dichromate capacity method, which used a Guohua HH-S oil bath to heat (Changzhou Guohua Electric Appliance Ltd., China), and the soil pH was measured with a PHS-3C pH meter (INESA Ltd., China; soil and water ratio of 2.5:1). The available nitrogen was diffused by alkali hydrolysis, and the available phosphorus and total phosphorus were determined by sodium hydroxide melt-molybdenum-antimony resistance colorimetry and $NaHCO_3$ extraction molybdenum-antimony resistance colorimetry, respectively, using a METASH UV-5500 ultraviolet–visible spectrophotometer (Shanghai Metash Instruments, China; *Li et al., 2019*). Then, the total potassium and available potassium were determined by sodium hydroxide melting flame photometry and $NH_4OAc$ leaching flame photometry, respectively, using the Xinyi 6400A flame photometer (Shanghai Xinyi Instrumentation Ltd., China; *Liu & Wang, 2020*).

### Soil aggregate stability index and carbon sequestration calculation

Each water-stabilized aggregate fraction refers to the weight of each particle size agglomerate as a percentage of the total dry weight of the sample. The MWD, GMD, D, and K were determined to acquire the soil stability indexes (*Cao et al., 2021*). The aggregate CPC was calculated to determine the carbon fixation capacity of the different aggregates (*González-Rosado et al., 2023*).

$$MWD = \sum_{i=1}^{n} x_i w_i \tag{1}$$

$$GMD = Exp\left(\sum_{i=1}^{n} w_i \ln x_i\right) \bigg/ \left(\sum_{i=1}^{n} w_i\right) \tag{2}$$

$$D = 3 - \log\left[\frac{m_{(i<x_i)}}{m_t}\right] \bigg/ \log\left[\frac{x_i}{x_{max}}\right] \tag{3}$$

$$K = 7.594\left\{0.0034 + 0.0405\exp\left[-\frac{1}{2}\left(\frac{lg^{GMD} + 1.659}{0.7101}\right)^2\right]\right\} \tag{4}$$

$$CPC = \frac{wc_i * w_i}{50} \tag{5}$$

where $x_i$ is the mean diameter of grade $i$ aggregate fraction (mm); $x_{max}$ is the average diameter (mm) of the maximum aggregate components; $w_i$ refers to the percentage content (%) of the aggregate of class $i$; $m_{(i<x_i)}$ is the dry weight (g) of aggregate size less than $i$; $m_t$ mean something of the initial total dry weight (g) of soil sample; $wc_i$ is the organic carbon content of particle $i$ (g kg$^{-1}$).

### Data analysis

In this study, SPSS 25.0 was used for the statistical analysis (*Li et al., 2015*). A one-way analysis of variance (with Duncan's multiple-range test) was used to compare the particle size distribution, MWD, GMD, K, D, SOC, and CPC of the aggregates at the $p = 0.05$ significance level (*Farhadi et al., 2023*). Origin 2023b was used to analyze the aggregate particle size distribution, aggregate stability, and aggregate-associated organic carbon using correlation analysis and principal component analysis (*Xu et al., 2023*).

## RESULTS

### Aggregate size distribution under the different vegetation restoration types

The distribution characteristics of the soil water stability aggregates under the diverse vegetation restoration modes were similar in each soil layer (from 0 to 40 cm; Fig. 2). At the depth of 0–40 cm, the distribution of the aggregates was significantly different between the different vegetation restoration modes, with >0.25 mm (>2 mm and 0.25–2 mm) as the dominant aggregate, and the mean value was 73.07%, which was higher than that of the control (FL; $P < 0.05$). Only the >2 mm aggregate components of the GL and SL were at a significantly higher proportion than those of the other vegetation types ($P < 0.05$). However, the <0.25 mm aggregate was the dominant size in the FL, and its proportion was significantly higher than that of the GL, GP, SL, and WL ($P < 0.05$). Except for the FL, the 0.25–2 mm aggregates were at the highest proportion in all the vegetation types, and the percentage of the 0.25–2 mm aggregates was in the following order: GP (51.37%) > WL (48.11%) > GL (46.87%) > SL (46.63%) > FL (28.78%), which was 43.97%, 40.18%, 38.59% and 38.28% higher than that of the FL. The proportion of the >2 mm aggregate
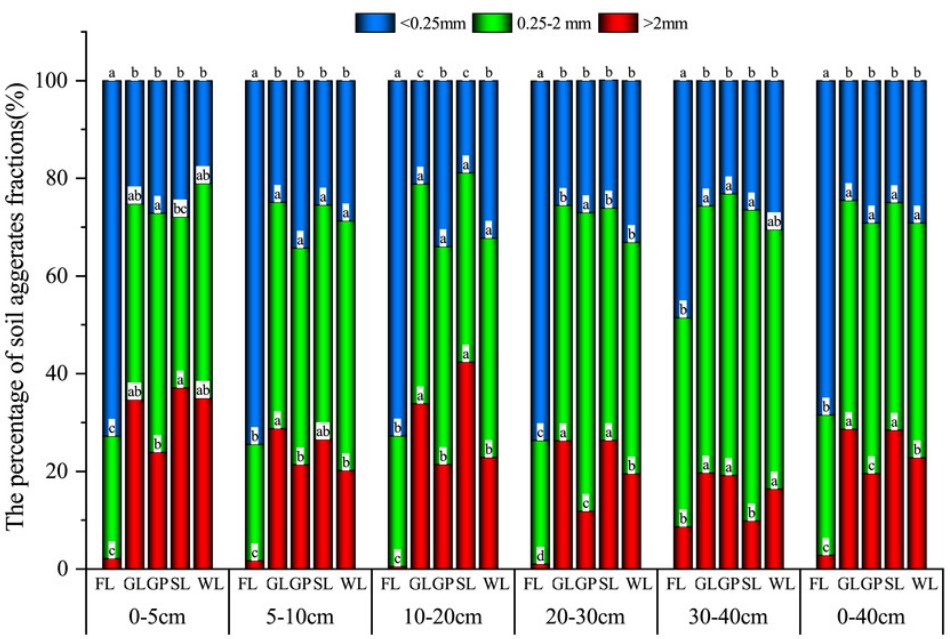

**Figure 2  Distribution characteristics of soil water stability aggregates under different vegetation restoration patterns.** Farmland (FL), grassland (GL), shrubland (SL), woodland (WL), and garden plots (GP). Different lowercase letters for the same soil aggregate size show the significant difference between different vegetation patterns ($P < 0.05$).

was in the order of GL (28.63%) > SL (28.42%) > WL (22.76%) > GP (19.52%) > FL (2.79%), and when compared with that of the FL, it increased by 85.73%, 90.20%, 90.27%, and 87.76%, respectively. Additionally, the >0.25 mm (>2 mm and 0.25–2 mm) aggregate fractions were in the order of GL (75.50%) > SL (75.05%) > GP (70.89%) > WL (70.87%) > FL (31.57%). Except for the FL, the proportion of 0.25–2 mm aggregates in the topsoil (0–5 cm, 5–10 cm, and 10–20 cm) was less than that in the lower soil (20–30 cm and 30–40 cm), and the percentage of >2 mm aggregates had a contrary pattern. While the content of the <0.25 mm fractions showed a slight variation in the 0–40 cm soil layer, the mean value was greater than that of the large aggregates. Therefore, the proportion of >0.25 mm aggregates also did not significantly change depending on the soil layer.

## Variation in the soil aggregate stability under different vegetation restoration types

The MWD and GMD of the four vegetation types in the 0–40 cm layer were higher than those of the FL, and their ranges were 0.93–0.72 and 0.58–0.96, respectively (Fig. 3). The MWD and GMD of the four vegetation types increased in the following order: GL > SL > GP > WL, which were 62.12%, 61.20%, 59.03%, 58.18% (MWD) and 64.80%, 63.49%, 61.06%, 59.26% (GMD) higher than those of the FL, respectively. With an increase in the soil depth, the MWD and GMD values of the WL and SL tended to decrease. The MWD and GMD values of the GL first increased and then decreased. Whereas those of

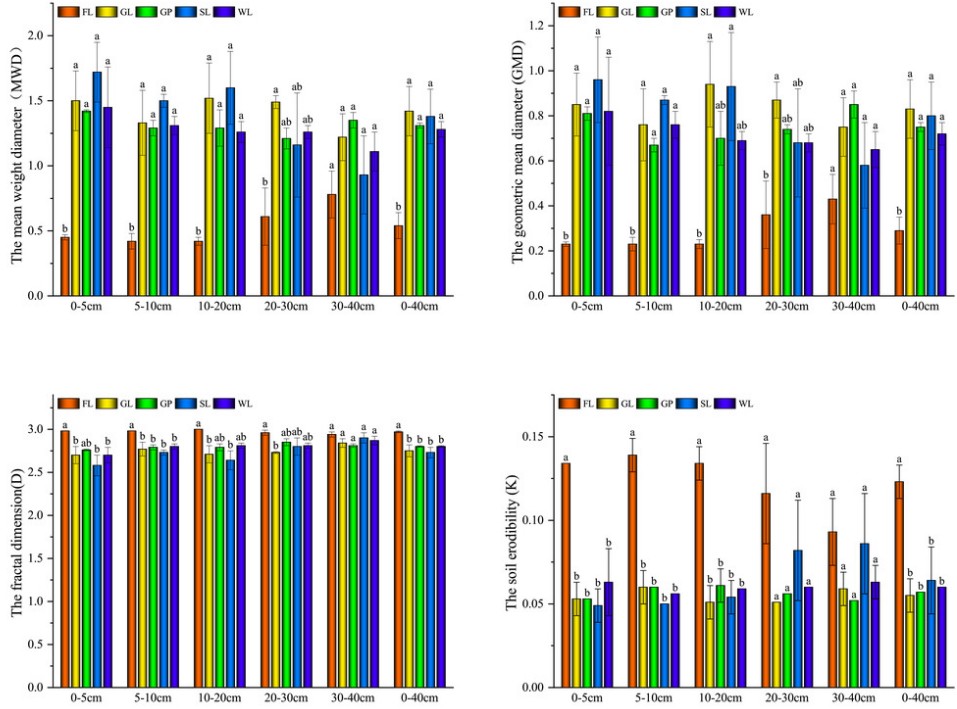

**Figure 3** **The aggregate stability under different vegetation restoration patterns.** Farmland (FL), grassland (GL), shrubland (SL), woodland (WL), and garden plots (GP). Different lowercase letters show the significant difference between different vegetation restoration patterns ($P < 0.05$).

the GP first decreased and then increased, and those of the FL increased continuously. In contrast, the D and K were the highest in the FL but there was no significant difference between the different vegetation types. Hence, the soil stability of the SL in the topsoil (0–20 cm) was greater than that of the GL in the underlying soil layers (20–40 cm). The four stability parameters of the different vegetation types were significantly higher than those of the FL ($P < 0.05$), and the differences in the surface soil layers (0–5 cm, 10 cm, and 10–20 cm; $P < 0.05$) gradually disappeared in the deeper soil layers (20–30 cm and 30–40 cm).

## The variation in the aggregate-associated soil organic carbon content for the different vegetation types

As shown in Fig. 4, the soil aggregate-associated SOC content in the GL, SL, GP, and WL was higher than that in the FL in the 0–40 cm soil layer. The soil aggregate-associated SOC content was in the following order: GL > SL > WL > GP > FL, where it was 48.57%, 40.02%, 40.20%, and 37.77% higher than that of the FL, respectively. When compared with the FL, the SOC content of the >2 mm, 0.25–2 mm, and <0.25 mm aggregates in the GL significantly increased by 46.13%, 39.51%, and 59.06%, respectively. The >2 mm and <0.25 mm aggregate organic carbon contents of the GP were higher than those of the FL at 39.86% and 49.42%, respectively. Additionally, the SL significantly increased the organic carbon content of the three aggregate size classes (>2 mm, 0.25–2 mm, and <0.25 mm),

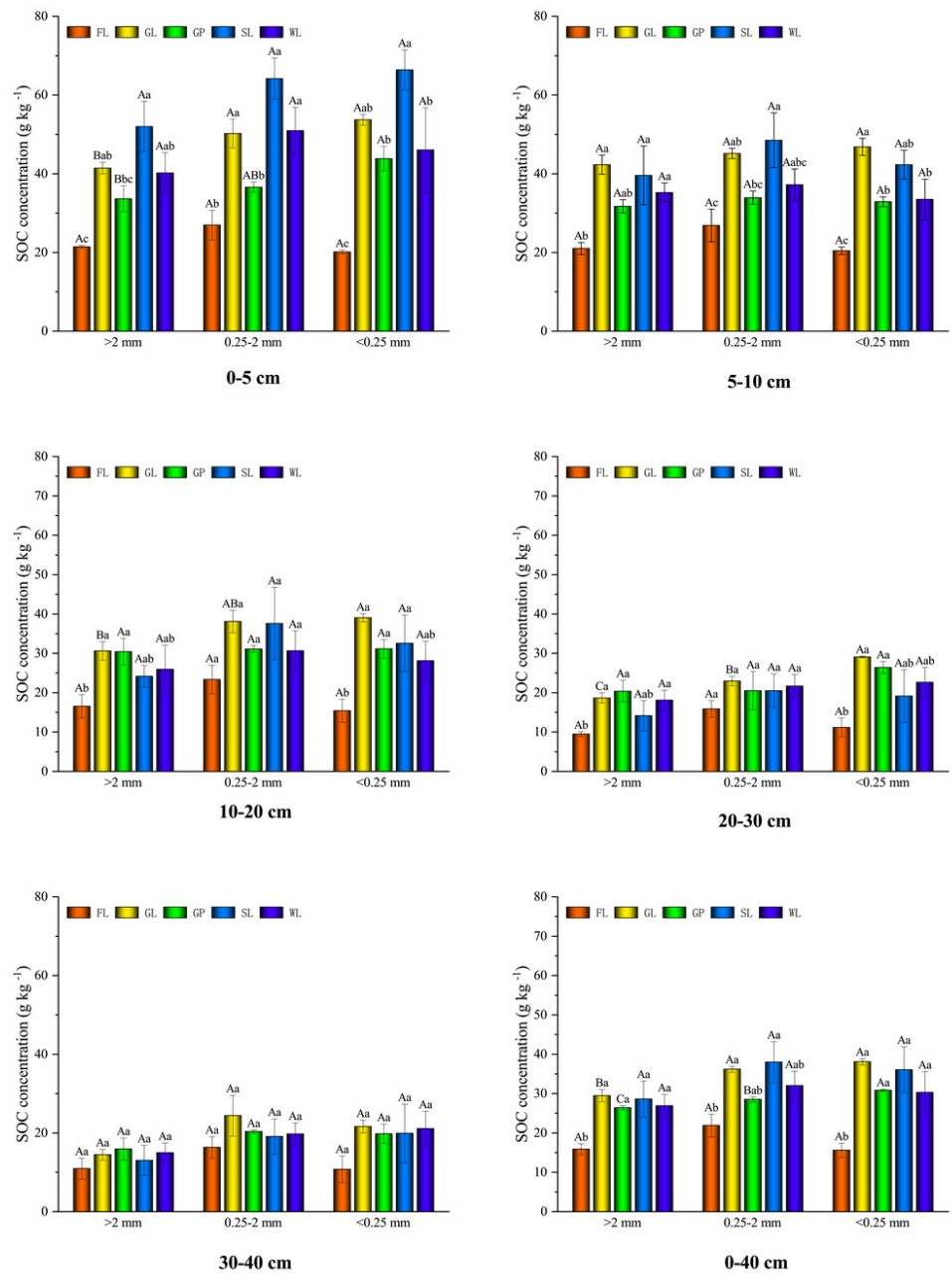

**Figure 4 The distribution of aggregate-associated carbon under different vegetationrestoration patterns.** Different lowercase letters for the identical soil aggregate size class indicate the significant difference between different vegetation types ($P < 0.05$). Different capital letters stand for differences in aggregate size under the same vegetation restoration patterns.

with the values being 44.44%, 42.41%, and 56.79%, respectively. The WL significantly increased the >2 mm and <0.25 mm SOC by 40.94% and 48.52%, respectively. Among the different vegetation types, the SOC of the GP was significantly different from that of the other vegetation types for the 0–5 cm and 5–10 cm soil layers ($P < 0.05$). The

aggregate organic carbon content was mainly distributed in the 0.25–2 mm and <0.25 mm aggregates, with mean values of 33.66 g kg$^{-1}$ and 33.80 g kg$^{-1}$, respectively, which were 17.29% and 17.63% higher, respectively, than that of the >2 mm aggregates (27.84 kg$^{-1}$). In the GL, the macro aggregates were significantly smaller than the meso-aggregates and micro-aggregates. In addition, significant differences existed between the macro aggregates, meso-aggregates, and micro-aggregates' aggregate-associated SOC in the SL, which was in the following order: micro-aggregates >meso-aggregates >macro aggregates. In the soil layer, there was a declining tendency in the aggregate-associated SOC content of all the vegetation types with the increase in the depth of the soil layer, and all the vegetation types were not significantly different from that of the FL at 30–40 cm.

## Response of the soil carbon preservation capacity to the different vegetation restoration patterns

Under the different vegetation restoration types, the soil total CPC was in the order of: GL > SL > WL > GP > FL in the 0–40 cm soil layer (Fig. 5). All the vegetation types significantly improved the CPC in the >2 mm and 0.25–2 mm aggregates, and it was significantly increased in the GL, SL, WL, and GP by 94.30%, 94.79%, 92.31%, and 91.87% (>2 mm aggregates) and 58.77%, 58.02%, 55.12%, and 52.38% (0.25–2 mm aggregates). In these aggregates, the 0.25–2 mm aggregates had the highest CPC, which ranged up to 15.25 g kg$^{-1}$, and the <0.25 mm aggregates had a mean value of 9.78 g kg$^{-1}$. The CPC of the 0.25–2 mm and <0.25 mm aggregates was 53.91% and 28.15% higher than that of the >2 mm aggregates, respectively. The CPC of the GL, SL, and WL were significantly larger than that of the macro aggregates ($P < 0.05$). Nevertheless, the GP showed significant differences in the different particle sizes, and they were in the following order: meso-aggregates > micro-aggregates > macro-aggregates ($P < 0.05$). The changes in the CPC of the different vegetation types in the soil layer showed that the CPC of the shallow soil layer was higher than that of the deeper soil layer. The CPC also showed a declining tendency with the increase in the depth of the soil layer. In these soil layers, the highest value was at 0–5 cm, and the mean value was 61.44 g kg$^{-1}$. In the topsoil, the lowest value was for the GP but in the deeper soil, the lowest value was for the SL.

As shown in Fig. 5B, the different vegetation restoration types improved the soil CPC but only that of the SL was significantly higher than that of the GP and WL at 0–5 cm. Moreover, the CPC of the SL and GL was significantly higher than that of the GP at 5–10 cm ($P < 0.05$).

## Pearson correlation analysis of the soil total carbon preservation capacity, aggregate-associated soil organic carbon content, soil aggregate proportion, and aggregate stability

In Fig. 6, the soil MWD and GMD were positively related to the >2 mm fraction, 0.25–2 mm fraction, >2 mm SOC, <0.25 mm SOC, SOC, >2 mm CPC, 2–0.25 mm CPC, and CPC ($P < 0.05$) but they were negatively correlated with the D, K, and <0.25 mm fraction ($P < 0.01$). Moreover, the D and K also had a negative relationship with the >0.25 mm fraction, SOC, >0.25 mm CPC, and CPC and a significant positive relationship with the <0.25 mm fraction ($P < 0.05$). The SOC had a significant positive relationship with the

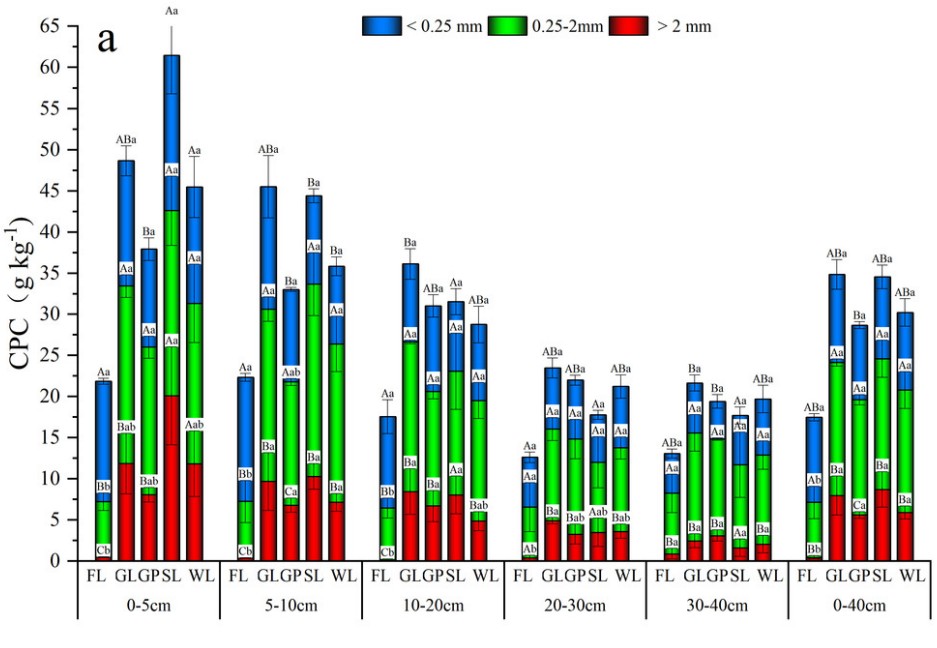

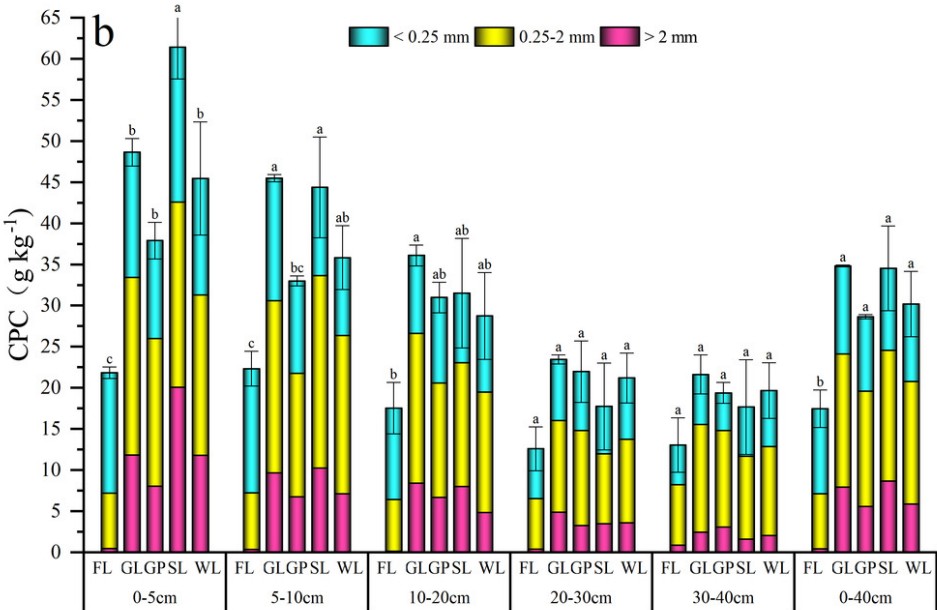

**Figure 5** **Carbon preservation capacity (A) and total carbon preservation capacity (B) in soil water stable aggregates under different vegetation restoration patterns.** Farmland (FL), grassland (GL), shrubland (SL), woodland (WL), and garden plots (GP). Different lowercase letters for the identical soil aggregate size stand for the significant difference between different vegetation restoration patterns ($P < 0.05$). Different capital letters correspond to differences in aggregate size under the same vegetation restoration patterns ($P < 0.05$).

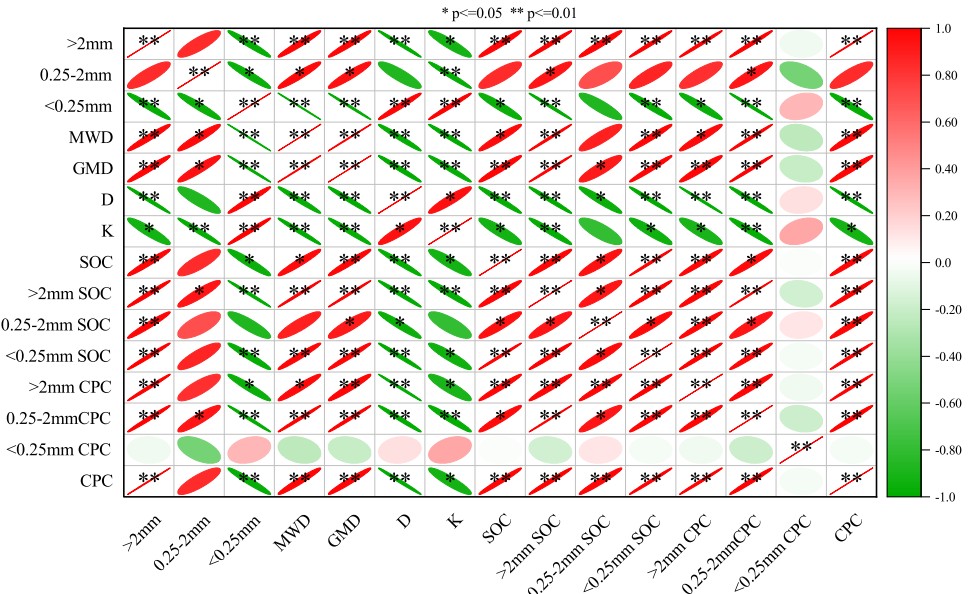

**Figure 6** **The person correlation between soil aggregate stability, carbon preservation capacity and carbon content of aggregates.** Mean weight diameter (MWD), fractal dimension (D), soil erodibility (K), geometric mean diameter (GMD), soil organic carbon (SOC) and carbon preservation capacity (CPC). * $p \leq 0.05$; ** $p \leq 0.01$.

>2 mm CPC, 0.25–2 mm CPC, >2 mm fraction, and aggregate-associated SOC ($P < 0.05$). Then, the >2 mm SOC was significantly correlated with the >0.25 mm fraction, 0.25–2 mm SOC, <0.25 mm SOC, >0.25 mm CPC, and CPC ($P < 0.05$). Furthermore, the 0.25–2 mm SOC was significantly correlated with the >2 mm fraction, <0.25 mm SOC, >0.25 mm CPC, and CPC ($P < 0.05$) and significantly negatively correlated with the K ($P < 0.05$). However, the <0.25 mm SOC had a significant positive association with the >2 mm fraction, >2 mm CPC, 0.25–2 mm CPC, and CPC ($P < 0.05$). There was a significant positive association between the >2 mm CPC, 0.25–2 mm CPC, and CPC, and they were all positively associated with the >2 mm fraction, aggregate-associated SOC, and SOC ($P < 0.05$). For the <0.25 mm CPC, the correlation between the soil aggregate stability parameters and the SOC content was not significant. In conclusion, the soil aggregate-associated SOC was significantly related to the aggregate stability ($P < 0.05$), and when the content of the >0.25 mm fraction was high, the soil MWD, GMD, and SOC reserves increased, while the D and K decreased.

## Principal component analysis of the soil aggregate proportion, aggregate stability, soil total carbon, aggregate-associated soil organic carbon, and soil organic carbon pools

As shown in Fig. 7, the first two principal components (PCs) accounted for 80.3% and 12.0%, respectively, and the cumulative variance contributed 92.3%, indicating that the first two principal components were the major contributors to the soil carbon reserves. When comparing the PC scores, PC1 explained about 6.7 times that of PC2. Additionally,

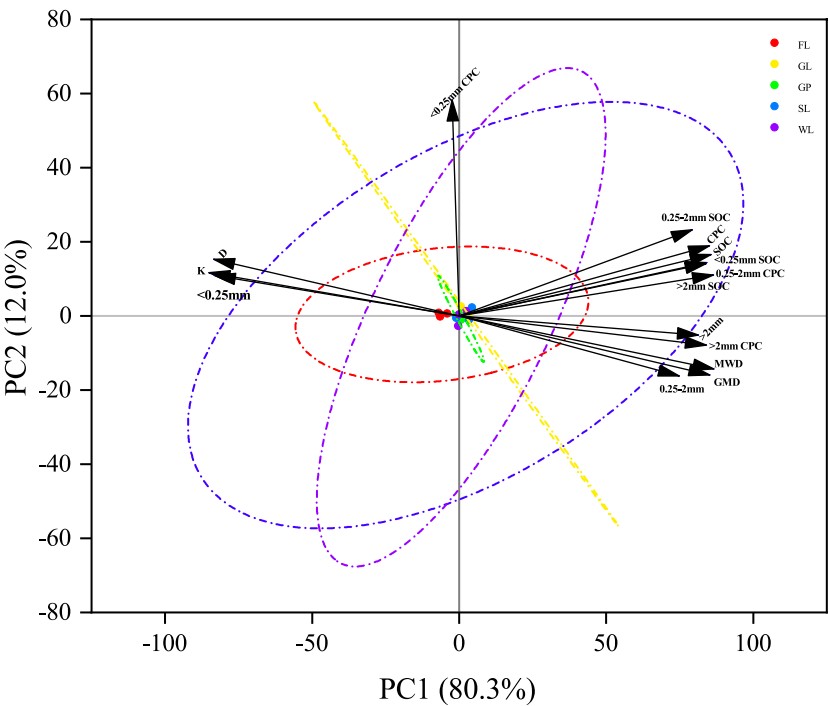

**Figure 7** **Principal component analysis of aggregate fractions, stability, aggregate associated soil organic content, total soil carbon and carbon preservation capacity.** Mean weight diameter (MWD), fractal dimension (D), soil erodibility (K), geometric mean diameter (GMD), soil organic carbon (SOC) and carbon preservation capacity (CPC).

the CPC, >2 mm CPC, 0.25–2 mm CPC, >2 mm fraction, 0.25–2 mm fraction, GMD, MWD, aggregate-associated SOC, and SOC were located on both sides of the PC1 axis, which indicates that soil carbon storage is associated with these indicators. The total CPC, >2 mm CPC, and 0.25–2 mm CPC increased with the GMD, MWD, >0.25 mm fraction, and >0.25 mm (>2 mm and 0.25–2 mm) aggregate-associated SOC increase. Therefore, improving aggregate stability can effectively improve the soil corrosion resistance and carbon sequestration of the soil aggregates.

# DISCUSSION

In this study, the relationship between CPC and aggregate stability was investigated. The results showed that vegetation restoration effectively improved soil carbon sequestration, and GL and SL were more effective for aggregate organic carbon sequestration than the other vegetation types.

## Soil aggregate proportion distribution and stability indexes under the different vegetation types

Previous studies have found that stabilized soil aggregates can influence the susceptibility of the aggregates to decomposition and segregation and the rate of decomposition of organic matter, which in turn affects soil crusting, erosion, and runoff, with consequences for the

soil quality (*Le Bissonnais, 2016*; *Wang et al., 2018*). When the micro-aggregate (<0.25 mm) proportion is too high it may impact the soil penetration function and microbial activity (*Zhang et al., 2019*; *Cao et al., 2021*). Therefore, different vegetation restoration patterns will affect vegetation growth. For instance, the surface litter will change the surface of the soil and the microbial environment, since microorganisms and nutrients are more readily released through the roots into the soil. Furthermore, the root and mycelium are the raw material for microbial decomposition activities and the synthesis of macro aggregate bonding material, which indirectly affect the soil aggregate composition distribution and stability (*Zhu, Shangguan & Deng, 2021*; *Bucka et al., 2021*).

This study found that vegetation restoration promoted the transformation of the micro-aggregates into large aggregates (Fig. 2). However, the <0.25 mm fraction was the dominant size class in the FL, and its macro aggregate and meso-aggregate contents were the lowest when compared with the other vegetation types (Fig. 2). This is because tillage not only has a mechanical impact on the aggregates but also brings subsoil to the surface, which makes the soil more susceptible to the influence of dry and wet cycles and accelerates the disintegration of the macro aggregates (*Denef et al., 2001*). This result is consistent with the findings of *Liu, Han & Zhang (2019)*, who found that at 0–20 cm in depth, the macro aggregate proportion of cropland was significantly smaller than that of abandoned land and land with natural plants. However, they also found that in the soil that was below 20 cm in depth, the difference between the three land uses was not significant, and the aggregate stability, organic carbon, and organic carbon content were elevated. Vegetation can promote the conversion of micro aggregates into large aggregates because the litter cover of the plants can increase the organic matter, promote macro aggregates, reduce the destruction of the surface soil quality and the formation of surface runoff, enhance rainwater infiltration, and provide more moisture for the plants, which can promote the growth of the plant roots (*Gyssels et al., 2005*; *Li et al., 2020*).

The MWD and GMD in the GL, SL, GP, and WL were higher than those in the FL, and the highest values were in the GL. Whereas the D and K of the different vegetation were smaller than those of the FL, which indicated that vegetation recovery improved soil stability (Fig. 3). This is because the roots of plants can adsorb soil particles onto their surface through intermolecular forces, thereby promoting the formation of soil aggregates and improving soil stability (*Wang et al., 2015*; *Wang & Zhang, 2017*; *Dong et al., 2022*). Soil stability is affected by soil property factors (*e.g.*, physicochemical properties, soil biology, nutrient availability, and water) and field management factors (*e.g.*, human activities; *Ilay & Kavdir, 2018*). Agricultural management measures, such as leaving the soil bare or plowing can lead to macro aggregate turnover and soil instability (*González-Rosado et al., 2022*). The topsoil had the best soil stability in the SL but the deeper soil had the best soil stability in the GL (Fig. 3). Root density and root depth are different between vegetation types, so these findings may be explained since the SL accumulates more litter in the surface soil and there is more root biomass accumulation and input in the underlying soil of the GL (*Sommer, Denich & Vlek, 2000*).

In terms of the soil layer, the >2 mm fraction, MWD, K, and GMD gradually declined with the increase in the soil layer depth (Figs. 2 and 3). This is likely due to the plant litter

and roots mainly accumulating on the soil surface, which indicates that the effect of the plants' recovery on the stability of the soil structure was more pronounced in the topsoil, and the litter and fine root biomass is an important source of soil C (*Lv et al., 2022*).

## The variation in the soil organic carbon content and aggregate-associated soil organic carbon pools under the different vegetation types

In this study, the four vegetation restoration patterns significantly increased the SOC content and carbon pools of the three aggregate sizes (Figs. 4 and 5). Their mean value was higher than that of the FL, which indicates that vegetation restoration can effectively improve soil carbon sequestration. Because vegetation restoration increases above- and below-ground vegetation biomass, the increase in vegetation coverage and the decrease of external interference reduce the soil erosion of the surface soil, improving the organic carbon content and sequestration (*Yu et al., 2022*).

In terms of the soil layer, the SOC content and reserves of all the vegetation restoration modes showed a declining tendency with the increase in the soil layer (Figs. 4 and 5). This may be because most of the root biomass is concentrated on the surface of the soil, and the surface soil comes into direct contact with litter and mosses, the surface soil microbial abundance is higher, and the microbes can decompose organic matter and promote C accumulation through *in vitro* modification and *in vivo* turnover (*Xiao et al., 2021*). The SOC content and CPC varied with the decrease in the soil aggregate size, which was similar to the change in the soil aggregate proportion (Figs. 2, 4 and 5) and supports the findings of *Fang et al. (2015)* and *Sommer, Denich & Vlek (2000)*. In this study, the aggregate-associated carbon was distributed largely in the >0.25 mm aggregate proportion (Fig. 4). This agreed with the findings of *Yu et al. (2022)*, who found that returning farmland to forest can significantly improve the SOC concentration, and the effect on the macro aggregates was more obvious than that on the other aggregate sizes. This may be due to the organic carbon, which is a cementing substance and can bind the micro-aggregates to the macro aggregates. Furthermore, the large aggregates generally contain more hyphae than the other aggregates, and the hyphae in the decomposed state can increase the concentration of organic carbon in the macro aggregates (*Tong et al., 2020*). In this study, the SOC and carbon pools decreased with the decrease in the aggregate particle size, which was similar to the response of the proportion of soil aggregates, and the SOC in the 0.25–2 mm and <0.25 mm fractions was slightly higher than that in the >2 mm aggregate fraction (Figs. 4 and 5). This may have been because the vegetation restoration period in this study was not long enough, and the SOC in the micro-aggregates had not been transferred into the macro aggregates. This is because SOC first accumulates in the micro-aggregates and is gradually transferred to the macro aggregates (*Sui et al., 2012*).

Generally, the SOC content is crucial for soil aggregate stability and composition, and the concentration of the organic carbon is closely correlated with the soil aggregates (*Sekaran, Sagar & Kumar, 2021*). This result is consistent with the findings of our study. In the correlation analysis, the SOC content and total CPC were significantly positively related to the >2 mm, >2 mm SOC, MWD, and GMD, while they were negatively related to the D

and K (Fig. 6). This indicated that the higher the SOC content within the >2 mm fraction, the higher the soil stability. In terms of the principal component analysis, the >0.25 mm fraction (>2 mm and 0.25–2 mm) and its SOC content were the main indicators of the soil carbon pools (Fig. 7), which is consistent with the findings of (Cao et al., 2021). This is because high organic carbon soil types can provide adhesive materials for the development of aggregates, which results in more organic carbon being fixed in the large aggregates (Arrouays, Vion & Kicin, 1995). The results showed that vegetation restoration effectively improved the sequestration of SOC and reduced soil erosion.

This study found that the SOC content of the GP in the 0–5 cm and 5–10 cm soil layers was significantly lower than that in the other vegetation types ($P < 0.05$). The SOC content of the SL was significantly higher than that of the GP and WL at 0–5 cm, and the SOC contents of the SL and GL were significantly higher than that of the GP at 5–10 cm ($P < 0.05$; Fig. 4). These findings may be because of the field management (such as farming) as the strong soil disturbance accelerates the SOC decomposition and destroys the soil aggregate structure and sparse surface vegetation causes more serious soil erosion, which is not conducive to the formation of soil aggregates and soil aggregate stability (Six et al., 2000). This can result in organic carbon loss.

In this study, vegetation restoration improved the soil stability and carbon sequestration in the karst areas. Among the different vegetation types, the SL and GL had significantly higher CPC in the soil surface than in the other vegetation types, which indicates the stronger potential of soil carbon storage for these two vegetation restoration types (Fig. 5). The CPC of the GP in the topsoil was slightly lower than that of the other vegetation types (Fig. 5). Therefore, when future vegetation restoration is conducted using GP, its management should be strengthened to maintain the sustainable development of the soil in this region. Consequently, based on these findings, to protect the carbon sequestration capacity of karst regions and avoid an increase in atmospheric $CO_2$, there is an urgent need to reduce the impacts of human activities and strengthen vegetation restoration in the region. However, there were a few limitations to this study. Firstly, we only studied limestone soils but soils are highly heterogeneous in karst areas, and, thus, subsequent studies should consider other types of soils. Secondly, the indicators were limited in this study. Therefore, future studies could also combine biological indicators (such as microorganisms, enzyme activity, and soil animals) to provide a more comprehensive understanding of the factors influencing soil carbon sequestration after vegetation restoration.

## CONCLUSIONS

Vegetation restoration is important for improving the soil quality in ecologically fragile areas, reducing soil erosion, and increasing the organic carbon content, which slows global warming. The results showed that the distribution of the aggregates in the different vegetation restoration modes was dominated by the >0.25 mm aggregates. At 0–40 cm in depth, the >2 mm aggregate fractions of the GL and SL were at a significantly higher proportion than those of the other vegetation types ($P < 0.05$). The MWD and GMD of all the vegetation restoration modes were higher than those in the FL. In addition, vegetation

restoration effectively increased the SOC content and CPC, which were mainly distributed in the 0.25–2 mm and <0.25 mm fractions, and their mean values were 33.66 g kg$^{-1}$ (SOC), 15.25 g kg$^{-1}$ (CPC), 33.80 g kg$^{-1}$ (SOC), and 9.78 g kg$^{-1}$ (CPC), respectively. Among the different vegetation types, the SOC of the GP was significantly different from that of the other vegetation types for the 0–5 cm and 5–10 cm soil layers ($P < 0.05$). In addition, the different vegetation restoration types improved the soil CPC but only that of the SL was significantly higher than that of the GP and WL at 0–5 cm. Additionally, the CPC of the SL and GL were significantly higher than that of the GP at 5–10 cm ($P < 0.05$). The total soil CPC and MWD, GMD, and >2 mm and 0.25–2 mm aggregates and their SOC content were significantly positively correlated ($P < 0.01$), and the CPC was significantly negatively correlated with the D and K ($P < 0.05$). This indicated that the higher the proportion of the >0.25 mm aggregates, the lower the soil erosion, the more stable the soil, and the higher the soil carbon sequestration capacity. Additionally, the >2 mm SOC, 2–0.25 mm SOC, >2 mm CPC, 0.25–2 mm CPC, MWD, and GMD all gradually declined with the increase in the soil depth, which may be because the plant litter and roots are mainly distributed on the surface layer of the soil. These results indicated that the potential of carbon sequestration and the effect of reducing the $CO_2$ in the SL and GL vegetation types is better than that in the other vegetation types, and the effect of GP is relatively poor. Therefore, GL and SL restoration patterns may be more suitable in this study area, the protection of the vegetation should be strengthened, and cultivated land should be converted into other land-use methods with higher vegetation coverage and less tillage intensity that can effectively improve the soil structure and soil fertility.

### Funding

Financial support was provided by the China Ministry of Agriculture and Rural Affairs Project (Z2023365) and National Natural Science Foundation of China (31460133). The funders had no role in study design, data collection and analysis, decision to publish, or preparation of the manuscript.

### Grant Disclosures

The following grant information was disclosed by the authors:
China Ministry of Agriculture and Rural Affairs Project: Z2023365.
National Natural Science Foundation of China: 31460133.

### Competing Interests

The authors declare there are no competing interests.

### Author Contributions

- Hui Yang performed the experiments, analyzed the data, prepared figures and/or tables, authored or reviewed drafts of the article, writing-original draft, and approved the final draft.

- Hui Long conceived and designed the experiments, performed the experiments, analyzed the data, prepared figures and/or tables, and approved the final draft.
- Xuemei Li performed the experiments, prepared figures and/or tables, and approved the final draft.
- Xiulong Luo performed the experiments, prepared figures and/or tables, and approved the final draft.
- Yuanhang Liao performed the experiments, authored or reviewed drafts of the article, and approved the final draft.
- Changmin Wang performed the experiments, authored or reviewed drafts of the article, and approved the final draft.
- Hua Cai performed the experiments, authored or reviewed drafts of the article, and approved the final draft.
- Yingge Shu conceived and designed the experiments, authored or reviewed drafts of the article, and approved the final draft.

## Data Availability

The raw measurements are available in the Supplementary File.

## Supplemental Information

Supplemental information for this article can be found online at http://dx.doi.org/10.7717/peerj.16699#supplemental-information.

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
