# Peer review of "Vegetation restoration improved aggregation stability and aggregated-associated carbon preservation in the karst areas of Guizhou Province, southwest China"

_PeerJ, doi:10.7717/peerj.16699_

## Round 0.1 · original submission · Major Revisions

Please submit the paper after incorporating all comments of reviewers along with a point-to-point rebuttal letter. The citations recommended by reviewers are not necessary to incorporate, which can only be considered in case of a need. The language of the paper needs attention before the next round of review. The introduction should be enhanced explaining the literature gap and the contribution of the present research.

**Language Note:** The Academic Editor has identified that the English language must be improved. PeerJ can provide language editing services - please contact us at copyediting@peerj.com for pricing (be sure to provide your manuscript number and title). Alternatively, you should make your own arrangements to improve the language quality and provide details in your response letter. – PeerJ Staff

Reviewer 1 ·

Basic reporting

Title
The title is clear and provides a concise overview of the research topic. It mentions the key elements of the study, including "vegetation restoration," "aggregate stability," and "aggregated-associated carbon preservation," which helps potential readers understand the focus of the research.
While the title mentions "karst areas in southwest China," it could be more specific in terms of the region or province within southwest China to give readers a better idea of the study's geographical scope.
Abstract
The abstract effectively introduces the importance of soil carbon and its relation to atmospheric carbon dioxide, setting the context for the study. However, it could benefit from a more concise statement of the research gap or question the study aims to address.
The abstract briefly mentions the research objects (vegetation types) and the methods used for data collection and analysis. However, it lacks a clear statement of the study's objectives or hypotheses. Readers would benefit from knowing what specific questions the study aimed to answer.
The abstract concludes by summarizing the positive effects of vegetation restoration on soil quality and carbon preservation. While this is a valuable conclusion, it could be enhanced by discussing the broader implications of the findings and potential applications in land management or environmental policy.
The abstract is quite long and could be more concise by focusing on the most critical findings and implications.
Introduction
The introduction is quite lengthy and contains a lot of information. While it's important to provide context, some sections could be condensed for clarity and conciseness. It might benefit from a more structured approach, with subheadings or paragraphs to help readers navigate the content more easily.
The introduction mentions the problems of rocky desertification in the karst area and the potential role of vegetation restoration but doesn't explicitly state the research gap or specific objectives of the study until the end. It would be helpful to clearly state the research questions or objectives earlier in the introduction to guide the reader's understanding.
The introduction sets up two hypotheses, which is good. However, these hypotheses could be more concise and directly tied to the research objectives. State them clearly and succinctly.
There are several grammatical errors and awkward phrasings in the text that need to be addressed for clarity and readability.
Materials and Methods
The section is generally well-organized and provides a clear overview of the experimental area, design, and methods. However, there are some areas where clarity can be improved and could be benefited from additional detail or references.
https://doi.org/10.1111/ppl.13759
https://doi.org/10.1016/j.scienta.2023.112369
https://doi.org/10.1002/jsfa.9977
https://doi.org/10.1080/01904167.2022.2056483



The introduction of the experimental area provides essential information about its location and climate. However, including a map or diagram of the area could help readers visualize the study site better.
The description of soil analysis methods is detailed and clear. However, it would be helpful to provide information on the equipment used (e.g., make and model of the pH meter, spectrophotometer for nutrient analysis) to ensure transparency and reproducibility.
Ensure that units of measurement are consistent throughout the section and match the conventions commonly used in scientific literature.
Explain any abbreviations or specialized terms upon first use to ensure readers understand them.
Check for any grammatical errors or awkward phrasings to ensure clarity and readability.
Results
This section is presented Good but it should be checked for English Grammar and editing because in many areas the sentence structure is not good and understandable.
Discussion
The authors discussed the results in a better way but justification for each result should be added in reference with the previous literatures.
Conclusion
The conclusions mention that certain soil properties, including SOC content and MWD, decline with soil depth. It would be beneficial to discuss the potential reasons behind this pattern and its implications for soil management or conservation.
The conclusions mention that "GL vegetation was the best restoration effect," but it would be helpful to clarify what is meant by "best restoration effect" and provide specific details or reasons why GL vegetation outperformed other types.
The section briefly touches on the benefits of vegetation restoration for soil quality and carbon sequestration. Expanding on the practical implications of these findings for land management or environmental policy would add depth to the conclusions.
Avoid repetition of information, such as stating that GL had the best restoration effect multiple times. Instead, summarize this information once in a clear and concise manner.

Experimental design

no

Validity of the findings

no

Additional comments

no

Reviewer 2 ·

Basic reporting

The language used in the provided research material is generally clear, unambiguous, and written in a professional tone. The descriptions, methodologies, results, and conclusions are presented in a structured and organized manner. However, there are a few instances where sentence structures are intricate, which might require careful reading to fully grasp the intended meaning. The research material follows a professional article structure commonly used in scientific papers. The hypotheses could be stated explicitly at the beginning of the research material. This would provide a clear framework for the reader to understand the specific questions the study aimed to address.

Experimental design

Research question is well-defined, relevant, and meaningful. It specifically examines the impact of various vegetation restoration patterns on soil properties and carbon sequestration in a karst area. This is a pertinent question, considering the ecological significance of soil health and carbon sequestration in such environments. The study also clearly states how it fills an identified knowledge gap. It emphasizes the need to understand the effects of different vegetation restoration strategies on soil aggregates, stability, and associated carbon content in karst landscapes. This addresses a gap in existing literature, as there is limited research on this topic in the context of karst areas. Research demonstrates a rigorous investigation conducted to a high technical and ethical standard. The methodology outlines a systematic approach, including detailed descriptions of the experimental area, sampling procedures, and analytical techniques. The use of established methods for soil analysis and aggregate separation adds credibility to the study. The ethical considerations, such as obtaining consent from local villagers and ensuring proper soil sampling protocols, indicate a conscientious approach to research ethics. The study also emphasizes the need for careful handling and transportation of soil samples to maintain their integrity. The soil sampling process is described in great detail, including the sampling depths, the "S" type sampling method, and precautions taken to maintain the integrity of the soil samples. Additionally, the method for aggregate separation and physicochemical index determination is outlined step by step, including the instruments and techniques used.

Validity of the findings

The research paper does not explicitly assess the impact and novelty of the study. While the Methods section is detailed and allows for meaningful replication, the paper does not specifically emphasize the importance of replication in contributing to the existing literature. It would be beneficial for the paper to include a section discussing the potential impact of the research findings and how they contribute to the broader scientific community. Additionally, highlighting the novelty of the study would provide context for readers to understand how this research advances current knowledge in the field. The conclusions in the research paper are adequately stated and are linked to the original research question. They are also supported by the results presented in the study. The paper effectively summarizes the findings and their implications in the context of the research objectives.

Additional comments

The research paper provides a comprehensive and detailed analysis of soil aggregate distribution and stability under different vegetation restoration patterns. The methodology is well-described, allowing for potential replication. The paper is written in clear and professional language throughout.
However, there could be more emphasis on the significance of the findings in the context of broader ecological and environmental implications. Additionally, while the paper is well-structured, some sections, especially in the Results and Discussion, could benefit from subheadings for better organization and clarity.

Annotated reviews are not available for download in order to protect the identity of reviewers who chose to remain anonymous.

---

## Round 0.2 · accepted · Accept

The paper is improved after revision and is accepted for publication.

Reviewer 1 ·

Basic reporting

Accepted

Experimental design

no

Validity of the findings

no

Additional comments

no